# Study of a QueryPNet Model for Accurate Detection and Segmentation of Goose Body Edge Contours

**DOI:** 10.3390/ani12192653

**Published:** 2022-10-02

**Authors:** Jiao Li, Houcheng Su, Xingze Zheng, Yixin Liu, Ruoran Zhou, Linghui Xu, Qinli Liu, Daixian Liu, Zhiling Wang, Xuliang Duan

**Affiliations:** 1College of Information Engineering, Sichuan Agricultural University, Ya’an 625000, China; 2Institute of Collaborative Innovation, University of Macau, Taipa, Macau 999077, China

**Keywords:** precision animal husbandry, computer vision, instance segmentation, target detection, neck module

## Abstract

**Simple Summary:**

Precision animal husbandry based on computer vision has developed promptly, especially in poultry farming. It is believed to improve animal welfare. To achieve the precise target detection and segmentation of geese, which can improve data acquisition, we newly built the world’s first goose instance segmentation dataset. Moreover, a high-precision detection and segmentation model was constructed, and the final mAP@0.5 of both target detection and segmentation reached 0.963. The evaluation of the model showed that the automated detection method proposed in this paper is feasible in a complex environment and can serve as a reference for the relevant development of the industry.

**Abstract:**

With the rapid development of computer vision, the application of computer vision to precision farming in animal husbandry is currently a hot research topic. Due to the scale of goose breeding continuing to expand, there are higher requirements for the efficiency of goose farming. To achieve precision animal husbandry and to avoid human influence on breeding, real-time automated monitoring methods have been used in this area. To be specific, on the basis of instance segmentation, the activities of individual geese are accurately detected, counted, and analyzed, which is effective for achieving traceability of the condition of the flock and reducing breeding costs. We trained QueryPNet, an advanced model, which could effectively perform segmentation and extraction of geese flock. Meanwhile, we proposed a novel neck module that improved the feature pyramid structure, making feature fusion more effective for both target detection and instance individual segmentation. At the same time, the number of model parameters was reduced by a rational design. This solution was tested on 639 datasets collected and labeled on specially created free-range goose farms. With the occlusion of vegetation and litters, the accuracies of the target detection and instance segmentation reached 0.963 (mAP@0.5) and 0.963 (mAP@0.5), respectively.

## 1. Introduction

With the rapid growth in the world’s population, the demand for meat and egg products with high nutritional value is increasing.

In 2020, 76.39 million tons of pork, cattle, sheep, and poultry meat were produced in China, of which 23.61 million tons of poultry meat were produced, an increase of 5.5% year-on-year, accounting for 30.9% of the total meat production. Goose farming is one of the important industries in poultry farming, and it can provide abundant egg and meat agricultural products. In 2020, global goose slaughter reached 740 million, an increase of 316 million compared to 2019 and an increase of 74.53% year-on-year [1].

In the process of large-scale livestock breeding, the risk of epidemics in animals increases due to the increase in breeding density, and the difficulty and cost of monitoring and management by hand increases due to the expansion of the breeding scale [2]. For this reason, realized intelligent precision farming can improve the poultry-rearing scale and product quality of farming, as well as the welfare and management of poultry farming to provide sustainable agricultural products [3]. Hence, how to visually monitor and control breeding while livestock farming has become an important topic in precision farming [4,5]. Typically, precision livestock farming often uses wearable devices for the precision farming of animals [6]. For example, pigs and cattle are identified in livestock farming through radio frequency technology (RFID) [7].

In recent years, due to improvement in computing power, deep learning has ballooned, thus bringing more solutions for computer vision. Computer vision has been gradually applied to people’s lives and production in various ways, such as face recognition, human flow detection, etc. [8]. Based on this, it is gradually being applied in animal husbandry farming. Non-invasive monitoring methods using sensors and cameras to acquire data and then processing the data through computer vision is a research hotspot in precision animal husbandry today [9]. The acquisition of livestock images using cameras and other means, followed by automated monitoring with the aid of computer vision, can result in substantial labor and equipment cost savings. Zheng Xingze et al. estimated the sex of sisal ducks through a two-stage detection method with target detection and a classification network and achieved an accuracy rate as high as 98.68% [10]. Lin Bin et al. conducted a study related to the estimation of fish pose using rotating target detection and a pose estimation algorithm [11]. Liao Jie et al. effectively classified the sound of pigs with TransformerCNN. The correct rate reached 96.05% [12].

Instance segmentation is a new and important branch of computer vision that has emerged in recent years and is also challenging. It requires not only detecting all objects in an image, but also accurately segmenting each instance. Kai-Ming He et al. proposed Mask R-CNN based on Faster R-CNN with only a small increase in overhead and won the best score in the COCO Challenge 2016 [13]. Daniel Bolya et al. proposed a simple fully convolutional instance segmentation model, YOLACT, which was able to guarantee 33.5 on Titan XP [14]. Xinlong Wang et al. proposed a simpler and more flexible instance segmentation framework, SOLO, by introducing the concept of “instance class” and avoiding the traditional strategy of detection followed by segmentation (e.g., mask R-CNN) [15]; this was followed by SOLOV2, which improved the instance mask representation scheme so that each instance in the image could be segmented dynamically without using bounding boxes for detection and reduced the overhead through novel matrix non-maximum suppression (NMS) [16]. Hao Chen et al. proposed BlendMask by effectively combining instance-level information with low-granularity semantic information, which improved the prediction of masks and was 20% faster than Mask R-CNN [17]. Yuxin Fang et al.’s QueryInst instance segmentation method driven by the parallel monitoring of dynamic masks exploited the intrinsic one-to-one correspondence among object queries at different stages, as well as one-to-one correspondence between mask RoI features and object queries at the same stage, and achieved the best performance among COCO, CityScapes, and Youtube VIS, and other tasks, obtaining excellent test results and, in particular, the best performance in video instance segmentation and struck a decent speed–accuracy trade-off [18].

Although instance segmentation provides more valuable segmentation detection results, few studies have applied it to agricultural farming due to its complexity. Jennifer Salau et al. applied Mask R-CNN to the farming of dairy cattle with a given IOU threshold of 0.5 for bounding box (0.91) and segmentation mask (0.85). Ahmad Sufril Azlan Mohamed et al. extracted individual contours of cattle in images using enhanced Mask R-CNN and obtained a mAP of 0.93 [19]. Johannes Brünger et al. followed a relatively new definition of panoramic segmentation and proposed a new instance segmentation network that obtained a 95% F1 score with 1000 hand-labeled images [20].

Instance segmentation can be effective for counting, behavioral detection, body size estimation, and automated monitoring of livestock individuals in agriculture. However, there are few example segmentation studies related to poultry species applied at present. To realize precision livestock farming for geese, this paper improves the feature fusion part, thus proposing a model that can be applied to the instance segmentation of geese flocks to achieve more accurate segmentation and more comprehensive information extraction, thereby allowing for comprehensive breeding information monitoring and analysis.

Concretely, the contributions consist of the following main points:We propose a novel neck module to obtain multiscale features of targets for fusion, shortening the path of feature fusion between high and low levels and making both the detection of targets and the segmentation of individual instances more effective.We construct a new and efficient query-based instance segmentation method by reducing the number of training parameters through a rational design and combining it with the neck module.We build a new goose dataset containing 639 instance segmentation images including 80 geese, which can be used as a reference for poultry instance segmentation research. The goose dataset comes from a meat goose free-range farm. The dataset images have both single, individual goose and dense geese activities, which are disturbed by natural factors, such as vegetation shading, non-goose animals, water bodies, and litter. Such datasets come from free-range production farming, which has a more complex background environment than captive breeding and can make the trained model more robust.

In this paper, the dataset collection work is elaborated on in Section 2, the model part is explained in detail in Section 3, the training and experimental results of this paper’s model are presented in Section 4, and the results and future research directions are summarized in Section 5.

## 2. Materials and Methods

### 2.1. Data Collection

Geese data were collected from a private meat goose farm in Jiaxing City, Zhejiang Province. The farm uses a free-range farming method and has several breeding sites with a single-site breeding population of around 80 geese, which can be slaughtered in around 70–80 days. Set near the coast, it has access to sufficient water for the fattening of the geese in a flexible stocking system. The free-range method of breeding gives the geese a more natural growing environment compared to the captive breeding method, so the raised geese have better quality meat. The data on geese obtained in this environment are also more informative.

The recording device used was the DJI pocket2, a sports camera released by DJI in 2020 with 1/1.7” CMOS and 64 million effective pixels, a lens FOV of 93° f/1.8, a lens equivalent focal length of 20 mm, and a maximum photo resolution of 9216 × 6912 pixels that supports up to 4K Ultra HD when recording video. To ensure that the data had the maximum processing space, the 4K60FPS mode was chosen for recording, and a total of 3.5 h of raw video data were captured by randomly sampling multiple geese in different locations and camera positions.

To ensure that the dataset had better representativeness, we sampled the original video data at 10 frames and randomly acquired image data, obtaining a total of 3247 datasets. After data preprocessing, we obtained 639 final datasets of acceptable quality and controlled the image size at 1920 × 1080 for data annotation. The annotation was performed by four colleagues in the lab who had experience in data annotation using labelme with the coco dataset format. The final partial dataset images and annotations are shown in Figure 1.

Considering the following tips, the goose dataset was a daunting task for the segmentation network.

Green, scientific, free-range farming methods are more complex compared to the narrow and homogeneous environment of captivity, with various vegetation, running water, and other shading factors; the dataset had strong interference, making the experiment more challenging.The background environment of the dataset had feed-feeding core areas, edge areas, etc. There were frequent change situations in goose location, as well as image balances of geese in sparse and dense distributions, which made our network design have stronger robustness and generalization ability.The existence of a high degree of similarity in appearance between goose bodies made it difficult for both the human eye and the network to distinguish between specific geese, making it difficult for later flock analysis, so improving segmentation accuracy was key.In the goose detection task, it was also a great challenge to detect individual goose instances in a complex environmental context.

Our goose dataset was, therefore, highly representative and could be effectively tested against the model.

### 2.2. Data Enhancement

The captured video dataset was converted into an image dataset. Then, we first eliminated the blurred and non-goose-containing images. To enrich the training dataset as much as possible, we performed data augmentation on the dataset before training by using various data enhancement methods, such as CutMix data enhancement, mosaic, four-way flip, and random rotation.

#### 2.2.1. CutMix Data Enhancement [21]

The use of regional dropout strategies enhances the performance of target detectors and dynamic masks, and such strategies can direct a model to focus on the less discriminative parts of a dataset, thus allowing the network to generalize better and have better localization capabilities. On the other hand, current regional dropout strategies remove informative pixels from training images by covering them with black pixels or patches of random noise. Such removal is undesirable, as it leads to information loss and inefficiency in the training process. Therefore, this paper used the CutMix enhancement strategy: cutting and pasting blocks in the training image, which made the live labels also mixed proportionally with the area of the blocks. Better data enhancement was achieved by efficiently utilizing the training pixels and preserving the regularization effect of region loss.

x∈ℝW×H×C and y represent the training images and labels, respectively. The goal of CutMix is to generate a new training sample x˜,y˜ by combining two training samples: xA,yA and xB,yB. The generated training samples x˜,y˜ are used to train the model with its original loss function. The merge operation is defined as the following equation:(1)x˜=M⊙xA+1−M⊙xB
(2)y˜=λyA+1−λyB

$M\in {\left\{0,1\right\}}^{W\times H}$ denotes the binary mask indicating the location of deletion and padding from the two images, and ⊙ is multiplied element-by-element. As in Mixup [22], the combined ratio λ between two data points is sampled from the beta distribution Beta (α, α). To sample the binary mask M, we first sampled the bounding box coordinates $B=\left(r_{x},r_{y},r_{w},r_{h}\right)$ that represented the cropping regions on $x_{A}$ and $x_{B}$. Area B in $x_{A}$ was removed and filled with patches cropped from B of $x_{B}$.

In our experiments, we sampled a rectangular mask M with an aspect ratio proportional to the original image. The frame coordinates were sampled uniformly in the following manner:(3)$$r_{x}∼\mathrm{Unif}\left(0,W\right),r_{w}=W\sqrt{1-\lambda }$$
(4)$$r_{y}∼\mathrm{Unif}\left(0,H\right),r_{h}=H\sqrt{1-\lambda }$$
such that the cropped area ratio $\frac{r_{w}r_{h}}{\mathrm{WH}}=1-\lambda$. For the cropping region, the binary mask $M\in {\left\{0,1\right\}}^{W\times H}$ is determined by filling the bounding box B with 0; otherwise it is 1.

#### 2.2.2. Mosaic

First, the goose dataset was grouped, 4 images were randomly taken out of each group, and operations such as random inversion and random distribution were performed to stitch the 4 images together into a new image. By repeating this operation, the corresponding mosaic data enhancement images were obtained, enriching the detection and segmentation datasets and, thus, improving the robustness of the model.

#### 2.2.3. Flip

The flipping transformation is a common method of data enhancement and includes horizontal flipping, vertical flipping, and diagonal flipping (horizontal and vertical flipping are used simultaneously). A horizontal flip is a 180-degree flip from left to right or right to left, and a vertical flip is a 180-degree flip from top to bottom or bottom to top. Horizontal and vertical flips are more commonly used, but diagonal flips can also be used depending on the actual target.

#### 2.2.4. Random Color (Color Jitter)

Color jitter is random transformation to change the brightness, contrast, exposure, saturation, and hue of an image within a certain range to simulate changes in the image under different lighting conditions in a real shot, making the model learn from different lighting conditions and improving its generalization ability. This data enhancement method was used in the target detection of YOLOv2 [23] and YOLOv3 [24]. Online data enhancement (including color dithering) is performed on the training data in each batch during the training process, firstly transforming the image into HSV color space; then randomly changing the exposure, saturation, and hue of the image in the HSV color space; and then transferring the transformed image to the RGB color space.

#### 2.2.5. Contrast Enhancement

For some images, the overall darkness or brightness of the image is due to a small range of gray values, i.e., low contrast. Contrast enhancement is widening the gray-scale range of an image, e.g., an image with a gray-scale distribution between [50, 150] raises its range to between [0, 255]. A gray-scale histogram is used to describe the number of pixels or the occupancy of each gray scale value in the image matrix. The horizontal coordinate is the range of gray-scale values, and the vertical coordinate is the number of times each gray-scale value appears in the image. In practice, by plotting the histogram of an image, it is possible to clearly determine the distribution of gray values and to distinguish between high and low contrast. For images with low contrast, algorithms can be used to enhance their contrast. Commonly used methods include linear transformation, gamma transformation, histogram regularization, global histogram equalization, local adaptive histogram equalization (adaptive histogram equalization with restricted contrast), etc.

**Linear Transformation:** This algorithm changes the contrast and brightness of an image by linear transformation. Let the input image be I and the output image be O, with width W and height H. I(r,c) represents the gray value of the rth row and cth column of I, and O(r,c) represents the gray value of the rth row and cth column of O. The calculation formula is as follows:(5)$$O\left(r,c\right)=a\times I\left(r,c\right)+b,0\leq r<H,0\leq c<W$$
where a affects the contrast of the output image, and b affects the brightness of the output image. The contrast is amplified when a > 1 and reduced when 0 < a < 1; the brightness is enhanced when b > 0 and reduced when b < 0; O is a copy of I when a = 1 and b = 0. Similarly, the segmented linear transform can make different gray value adjustments in different gray value ranges to better suit the needs of image enhancement.

**Histogram regularization:** The parameters of the linear transformation need to be chosen reasonably according to different applications, as well as the information of the graph itself, and may need to be tested several times. Histogram regularization can automatically select a and b based on the current image situation. Let the input image be I, the output image be O, while the width is W, and the height is H. I(r,c) represents the gray value of the rth row and cth column of I. The minimum gray value of I is recorded as Imin, and the maximum gray value is recorded as Imax, and O(r,c) represents the gray value of the rth row and cth column of O. The minimum gray value of O is recorded as Omin, and the maximum gray value is recorded as Omax. To make the gray value range of O [Omin, Omax], the following mapping is performed:(6)$$a=\frac{\mathrm{Omax}-\mathrm{Omin}}{\mathrm{Imax}-\mathrm{Imin}},b=\mathrm{Omin}-\frac{\mathrm{Omax}-\mathrm{Omin}}{\mathrm{Imax}-\mathrm{Imin}}\times \mathrm{Imin}$$
(7)$$O\left(r,c\right)=\frac{\mathrm{Omax}-\mathrm{Omin}}{\mathrm{Imax}-\mathrm{Imin}}\left(I\left(r,c\right)-\mathrm{Imin}\right)+\mathrm{Omin}$$
(8)0≤r<H,0≤c<W

**Gamma transform:** The gamma transform is a nonlinear transform. Let the input image be I and the output image be O, with width W and height H. I(r,c) represents the gray value of the rth row and cth column of I, and O(r,c) represents the gray value of the rth row and cth column of O. The gray-scale values are first normalized to the range of [0, 1], and then calculated by the following equation:(9)$$O\left(r,c\right)=I{\left(r,c\right)}^{\gamma }$$
(10)0≤r<H,0≤c<W
where the image is constant at γ = 1, the contrast increases at 0 < γ < 1, and the contrast decreases at γ > 1.

**Global histogram equalization:** Gamma transform has a better effect in improving contrast, but the gamma value needs to be adjusted manually. Global histogram equalization uses the histogram of an image to automatically adjust the image contrast. Let the input image be I and the output image be O, with width W and height H. I(r,c) represents the gray value of the rth row and cth column of I, and O(r,c) represents the gray value of the rth row and cth column of O. ${\mathrm{hist}}_{I}$ represents the gray-scale histogram of I, ${\mathrm{hist}}_{I}\left(K\right)$ represents the number of pixels whose gray-scale value of I is equal to k, ${\mathrm{hist}}_{o}$ represents the gray-scale histogram of O, and ${\mathrm{hist}}_{o}\left(K\right)$ represents the number of pixels whose gray-scale value of O is equal to k, k ∈ [0, 255]. Global histogram equalization is a change to I such that the ${\mathrm{hist}}_{o}$ of O is equal to each gray value pixel point, i.e.:(11)$${\mathrm{hist}}_{o}\left(K\right)\approx \frac{H\times W}{256}$$

Then, for any gray value p (0 ≤ p < 255), it is always possible to find a gray value q (0 ≤ q < 255), such that:(12)$$\overset{p}{\underset{k=0}{\sum }}{\mathrm{hist}}_{I}\left(k\right)=\overset{q}{\underset{k=0}{\sum }}{\mathrm{hist}}_{O}\left(k\right)$$

$\overset{p}{\underset{k=0}{\sum }}{\mathrm{hist}}_{I}\left(k\right)$ and $\overset{q}{\underset{k=0}{\sum }}{\mathrm{hist}}_{O}\left(k\right)$ are called the cumulative histograms of I and O, respectively. Since ${\mathrm{hist}}_{o}\left(K\right)\approx \frac{H*W}{256}$, the following can be obtained:(13)$$\overset{p}{\underset{k=0}{\sum }}{\mathrm{hist}}_{I}\left(k\right)=\left(q+1\right)\frac{H\times W}{256}$$
(14)$$O\left(r,c\right)=\frac{\sum_{k=0}^{I\left(r,c\right)}{\mathrm{hist}}_{I}\left(k\right)}{H\times W}\times 256-1$$

**Local Adaptive Histogram Equalization:** While global histogram equalization is effective in improving contrast, it may also allow noise to be amplified. To solve this problem, local adaptive histogram equalization has been proposed. Local adaptive histogram equalization first divides an image into non-overlapping blocks of regions and then performs histogram equalization on each block separately. Obviously, without the influence of noise, the gray-scale histogram of each small region is limited to a small range of gray-scale values, but if there is noise influence, the noise is amplified after performing histogram equalization for each segmented block of regions. In general, each histogram can usually be represented by a column vector, and each value inside the column vector is a bin; for example, if a column vector has 50 elements, then it means there are 50 bins. Noise can be avoided by limiting the contrast, i.e., if a bin in the histogram exceeds the limit contrast set in advance, the excess is cropped and distributed evenly to other bins.

#### 2.2.6. Rotate

Rotate means to rotate the original image at different angles and has two cases: a random-angle rotation and a fixed-angle rotation. When the rotation angle is a multiple of 90 degrees, the size of the image does not change. Otherwise, the image is the size of an inner rectangle, and black borders appear.

#### 2.2.7. Center Clipping and Random Clipping

In image recognition tasks, clipping is a common method of data enhancement that allows areas of an image to be clipped while preserving the scale of the original image. Cropping can be achieved by intercepting an array of images using NumPy. There are three main types of cropping: center cropping, corner cropping, and random cropping. In this paper, center cropping and random cropping were used.

The various effects after each of the data enhancement operations are shown in Figure 2.

The final fusion effect of data enhancement is obtained in Figure 3:

### 2.3. Method

Our research focused on the identification and accurate segmentation of individual goose instances from complex backgrounds, enabling the fine extraction of contour features and facilitating group counting. This is a typical instance segmentation task and extension. In this paper, we attempted to use a query-based network model for goose instance segmentation, which was performed by combining the two subtasks of target detection (individual goose classification and localization) and semantic segmentation (identification of goose pixels) in one.

#### 2.3.1. QueryInst Network

QueryInst (Instances as Queries) is a query-based end-to-end instance segmentation method consisting of a query-based target detector and six dynamic masks driven by parallel supervision. The algorithm primarily exploits the one-to-one correspondence inherent in target queries across different stages, as well as the one-to-one correspondence between masked RoI features and target queries in the same stage. This correspondence exists in all query-based frameworks, independent of the specific instantiation and application. The R-CNN head of QueryInst contains 6 stages in parallel. The mask head is trained by minimizing dice loss [25]. The QueryInst model trained with ResNet-50 [26,27] as the backbone. The dynamic head architecture of QueryInst is shown in Figure 4.


**Query-based Object Detector**


QueryInst can be built on any multistage query-based object detector but is instantiated with Sparse R-CNN [28] as default, which has six query stages. The target detection implementation formula for geese is as follows:(15)$$\begin{array}{cc} & x_{t}^{\mathrm{box}}\leftarrow P^{\mathrm{box}}\left(x^{\mathrm{FPN}},b_{t-1}\right)\\ & q_{t-1}^{*}\leftarrow {\mathrm{MSA}}_{t}\left(q_{t-1}\right)\\ & x_{t}^{\mathrm{box}*},q_{t}\leftarrow {\mathrm{DynConv}}_{t}^{\mathrm{box}}\left(x_{t}^{\mathrm{box}},q_{t-1}^{*}\right)\\ & b_{t}\leftarrow B_{t}\left(x_{t}^{\mathrm{box}*}\right) \end{array}$$
where $q\in R^{N\times d}$ represents an object query. N and d represent the length (number) and dimension of query q, respectively. In the t stage, the pooling operator $P^{\mathrm{box}}$ extracts the current stage bounding box features $x_{t}^{\mathrm{box}}$ from the FPN features, guided by the $x^{\mathrm{FPN}}$ bounding box predictions of the previous stage $b_{t-1}$. A multihead self-attention module ${\mathrm{MSA}}_{t}$ is applied to the input query $q_{t-1}$ to obtain the transformed query $q_{t-1}^{*}$. Then, a box dynamic convolution module ${\mathrm{DynConv}}_{t}^{\mathrm{box}}$ takes the $x_{t}^{\mathrm{box}}$ sum $q_{t-1}^{*}$ as input and $q_{t-1}^{*}$ augments it by reading $x_{t}^{\mathrm{box}}$ while generating for the next stage $q_{t}$. Finally, the augmented bounding box features $x_{t}^{\mathrm{box}*}$ are fed into the box prediction branch $B_{t}$ for current bounding box prediction $b_{t}$.


**Dynamic Mask Head**


A query-based instance segmentation framework was implemented with a parallel supervision-driven dynamic mask head. The dynamic mask head at stage t consisted of a dynamic mask convolution module DynConvmask, followed by a vanilla mask head. The mask generation pipeline was reformulated as follows:(16)$$\begin{array}{cc} & x_{t}^{\mathrm{mask}}\leftarrow P^{\mathrm{mask}}\left(x^{\mathrm{FPN}},b_{t}\right)\\ & x_{t}^{\mathrm{mask}*}\leftarrow {\mathrm{DynConv}}_{t}^{\mathrm{mask}}\left(x_{t}^{\mathrm{mask}},q_{t-1}^{*}\right)\\ & m_{t}\leftarrow M_{t}\left(x_{t}^{\mathrm{mask}*}\right) \end{array}$$

The communication and coordination of object detection and instance segmentation were realized with dynamic mask headers.

#### 2.3.2. Model Architecture—QueryPNet


**Neck Design**


To enhance the propagation of information flow in the instance segmentation framework, we chose to use path aggregation networks in our model. High-level feature maps with rich segmentation information were used as one particular input for better performance.

Each building block obtained a higher-resolution feature map Ni and a coarser map P_i+1_ through lateral connections and generated a new feature map N_i+1_. Each feature map N_i_ was first passed through a 3 × 3 convolutional layer with a stride of 2 to reduce the spatial size. The feature map P_i+1_ and each element of the down-sampling map were then summed through lateral connections. The fused feature maps were then processed by another 3×3 convolutional layer to generate N_i+1_ for subsequent subnetworks. This was an iterative process. In these building blocks, we always used channel 256 of the feature map. All convolutional layers were followed by a ReLU. Then, the feature grids for each level were pooled from the new feature maps (i.e., {N_1_, N_2_, N_3_, N_4_}).

The implementation of the neck module in this paper was as follows, as shown in Figure 5:The information path was shortened, and the feature pyramid was enhanced with the precise localization signals present in the lower layers. The resulting high-level feature maps were then additionally processed using a bottom-up path enhancement method.Through the adaptive feature pool, all the features of each level were aggregated, and the features of the highest level were distributed to the same N_5_ levels obtained by the bottom-up path enhancement.To capture different views of each task, our model used tiny, fully connected layers to enhance the predictions. For the mask part, this layer had complementary properties to the FCN originally used by Mask R-CNN, and by fusing predictions from these two views, the information diversity increased and a better-quality mask was generated, while for the target in the detection part, a better-quality box could be generated.


**Proposed region generation and RoIAlign operation**


The obtained feature maps were sent to RPN [29], where the tens of thousands of candidate predictors in the region proposal network were no longer used. This paper chose to use 100 sparse proposals. This portion of sparse proposals was used as proposals to extract the regional features of the geese through RoIAlign. These proposal boxes were statistics of potential goose body locations in the images, which were only rough representations of goose targets, lacking many informative details, such as pose, shape, contour integrity, etc. Therefore, we set 256 high-dimensional proposal features (proposal_feature) to encode rich instance features. After that, a series of bounding boxes could be obtained, and for a case where multiple bounding boxes overlapped each other, non-maximum suppression (NMS) [30] was reasonably used to obtain bounding boxes with higher foreground scores, which were passed to the next stage.

In the backpropagation of the RoIAlign layer, $i^{*}\left(r,j\right)$ was the coordinate position of a floating-point number (the sample point calculated during forward propagation). In the feature map before pooling, the abscissa and ordinate of each point were $i^{*}\left(r,j\right)$ and less than 1, the corresponding point should be accepted.

The gradient of the RoIAlign layer was as follows:(17)$$\{\frac{\partial L}{\partial x_{i}}=\underset{r}{\sum }\;\underset{j}{\sum }[d\left(i,i^{*}\left(r,j\right)\right)<1]\left(1-△h\right)\left(1-△w\right)\frac{\partial L}{\partial y_{r,j}}$$
where $d\left(\cdot \right)$ represents the distance between two points, and △h and △w represent the difference between $x_{i}$ and $x_{i*}\left(r,j\right)$. Through the RoIAlign process, the extracted features were correctly aligned with the input image, which avoided losing the information of the original feature map in the process. The intermediate process was not quantized to ensure maximum information integrity, and it solved the problem of defining the corresponding region between the region proposal and the feature map. The problem of subpixel misalignment when defining the corresponding region between the region proposal and the feature map was solved, resulting in more accurate pixel segmentation. Especially for small feature maps, more accurate and complete information could be obtained.


**Goose target detection and instance segmentation**


This paper used 5 target detection heads and 5 dynamic mask heads, which could reduce the number of training parameters and optimize performance to a certain extent. The features obtained by RoIAlign used bbox_head to implement goose bounding box regression and mask_head to predict goose segmentation masks (goose body regions). For network training, the loss function represented the difference between the predicted value and the true value. It played an important role in the training of the goose segmentation model. For the loss function design of the two subtasks, we used CIoU loss [31] for bbox_head, which was also an adjustment to the original model, and dice loss for mask_head loss.

For CIoU loss, the implementation was as follows:(18)$$R_{\mathrm{CIoU}}=\frac{\rho^{2}\left(b,b^{\mathrm{gt}}\right)}{c^{2}}+\alpha v,$$
where α is a positive trade-off parameter, and v measures the consistency of following aspect ratio:(19)$$v=\frac{4}{\pi^{2}}{\left(\arctan\frac{w^{\mathrm{gt}}}{h^{\mathrm{gt}}}-\arctan\frac{w}{h}\right)}^{2}$$

Then, the loss function can be defined as:(20)$$L_{\mathrm{CIoU}}=1-\mathrm{IoU}+\frac{\rho^{2}\left(b,b^{\mathrm{gt}}\right)}{c^{2}}+\alpha v$$
and the trade-off parameter α is defined as:(21)$$\alpha =\frac{v}{\left(1-\mathrm{IoU}\right)+v}$$

Overlapping region factors were given higher priority in regression, especially for non-overlapping cases.

Finally, the optimization of CIoU loss was the same as that of DIoU loss, but the relative gradients were different.
(22)$$\frac{\partial v}{\partial w}=\frac{8}{\pi^{2}}\left(\arctan\frac{w^{\mathrm{gt}}}{h^{\mathrm{gt}}}-\arctan\frac{w}{h}\right)\times \frac{h}{w^{2}+h^{2}}$$
(23)$$\frac{\partial v}{\partial h}=-\frac{8}{\pi^{2}}\left(\arctan\frac{w^{\mathrm{gt}}}{h^{\mathrm{gt}}}-\arctan\frac{w}{h}\right)\times \frac{w}{w^{2}+h^{2}}$$

For cases $w^{2}+h^{2}$ in the range of [0, 1], the domination $w^{2}+h^{2}$ is usually a small value, which is likely to produce exploding gradients. Therefore, in specific implementation, in order to stabilize the convergence, the dominator is simply removed $w^{2}+h^{2}$, the step size $\frac{1}{w^{2}+h^{2}}$ is replaced by 1, and the gradient direction remains unchanged.

The dice loss is a loss function proposed based on the dice coefficient, which is calculated by the following formula:(24)$$D=\frac{2\sum_{i}^{N}p_{i}g_{i}}{\sum_{i}^{N}p_{i}^{2}+\sum_{i}^{N}g_{i}^{2}}$$
where the sums run over the N voxels of the predicted binary segmentation volume $p_{i}$∈ P and the ground truth binary volume $g_{i}$∈ G. This formulation of dice can be differentiated, yielding a gradient computed with respect to the j−th voxel of the prediction.
(25)$$\frac{\partial D}{\partial p_{j}}=2\left[\frac{g_{j}\left(\sum_{i}^{N}p_{i}^{2}+\sum_{i}^{N}g_{i}^{2}\right)-2p_{j}\left(\sum_{i}^{N}p_{i}g_{i}\right)}{{\left(\sum_{i}^{N}p_{i}^{2}+\sum_{i}^{N}g_{i}^{2}\right)}^{2}}\right]$$

The imbalance between foreground and background pixels was dealt with in the above way.

Figure 6 shows the main architecture of our model. After the data enhancement operation, the data were sent to the ResNet backbone to extract the features richly. To better utilize the features extracted by the backbone, the innovated PANet was used. Additionally, we utilized a parallel detection method, allowing the target detection head and the dynamic mask head to detect and segment data at the same time. Moreover, this part adopted a multihead attention mechanism to extend the ability of both detection and segmentation. Five pairs of parallel detection heads were used in this paper.

## 3. Results

To improve the training effect, a round of data augmentation was performed on the dataset first. The following data augmentation methods were applied to the dataset, as shown in Table 1.

### 3.1. Experimental Setup

A total of 639 high-quality datasets with pixels of 1920×1080 were selected, and the datasets were randomly scrambled and divided into an 8:1:1 ratio of training set, test set, and validation set, respectively.

The Pytorch framework was chosen to build, optimize, and evaluate the model designed for goose instance segmentation. Table 2 below is the basic equipment information of the software and hardware used in this paper.


**Training Setup**


We set the initial learning rate of the model to 0.00025 and used the AdamW optimizer with a weight decay rate of 0.0001. Meanwhile, due to AdamW’s rapid convergence, we set epochs to 120 to ensure effective convergence of the validation set.


**Inference**


Given an input image, the model directly output the top 100 bounding box predictions, along and their scores and corresponding instance masks, without further postprocessing. For inference, we used the final stage mask as prediction and ignored all parallel dynamic tasks in the intermediate stages. The reported inference speed was measured using a single TITAN V GPU with input resized to be 800 on the short side and less than or equal to 1333 on the long side.

### 3.2. Performance Evaluation Metrics

To fully verify the accuracy of the model, we conducted a comprehensive and objective evaluation of our model from the following metrics.


**IoU**


This was the ratio of the intersection and union of the target predicted and ground-truth boxes. The ratio was true positives (TPs) divided by TP, the sum of false positives (FPs) and false negatives (FN). FN meant the prediction was negative, but the flagged result was positive; FP was a negative situation, while for TP, the prediction was positive. In fact, this was also a positive example that the prediction was correct, where p^ij^ represented the number of real values and was predicted to be j, and k + 1 was the number of classes (including background). p^ii^ was the number of values correctly predicted, and p_ij and p_ji represented FP and FN, respectively. The formula for calculating IoU was as follows:(26)$$\mathrm{IoU}=\frac{1}{k+1}\overset{k}{\underset{i=0}{\sum }}\frac{p_{\mathrm{ii}}}{\sum_{j=0}^{k}p_{\mathrm{ij}}+\sum_{j=0}^{k}p_{\mathrm{ji}}-p_{\mathrm{ii}}}$$

When IoU was a different threshold, considering the difference in the size of the goose body in the image, the evaluation indicators in Table 3 were as follows.

### 3.3. Comparison with State-of-the-Art Methods and Results

This paper mainly explored high-precision performance networks that could achieve goose body detection and segmentation. Therefore, referring to various instance segmentation networks, we chose the mainstream networks in recent years (Mask R-CNN, YOLACT, PointRend, SOLO, SOLOv2, BlendMask, QueryInst, and SparseInst) for performance comparison with our QueryPNet. For different task networks, the AP values under different thresholds for the validation set are shown in Table 4 and Table 5, and the diagram is shown in Figure 7.

The main purpose of this paper was to study a high-precision target detection and segmentation network for geese that captured individual geese and achieved accurate outline extraction of goose instances to facilitate the later study of goose behavior, body size, count, etc.

After the experiments and analyses of the above results table, we chose to perform performance enhancement improvements on the query-based QueryInst network and, ultimately, obtained QueryPNet. Comparing the results of different task networks, the highest accuracy was achieved in both the detection of geese and the segmentation of goose instances, which met our research purposes.

The effect images of other models and of the QueryPNet model are shown in Figure 8 and Figure 9. In the frame selection detection of the target, Mask R -CNN, PointRend, SOLOv2, QueryInst, SparseInst, etc. had omissions, false detections, etc. Among these, the YOLACT model detected and segmented mAP results on our dataset that were significantly lower, and the segmentation effect was less than ideal, failing to achieve the effect of practical applications. The QueryInst model could segment a relatively complete goose body area, but the precise extraction of goose body contours and edge features needed to be improved. From the visual analysis results in Figure 8 and Figure 9, it can be seen that, after using our QueryPNet model, more real and accurate details could be generated, more instance information could be carried, and the misjudged pixels were greatly reduced. The segmented goose body area, especially the area near the edge of the goose body and the legs, was obviously more in line with the real goose outline on the original image, while the feature areas segmented by other models had results of failures, such as transgressions and lack of division.

In order to analyze the comprehensive performance of QueryPNet, we compared it with QueryInst. The results are shown in Table 6.

According to the analysis in Table 6, QueryPNet achieved subtasks with 2% and 0.5% higher mAP values for detection and segmentation, respectively, than the original model, and the performance in other aspects was also significantly improved. The complexity of the improved model and the cost of training parameters were significantly reduced, which were 16.89% and 19.43% lower than the original, respectively, and it received a 52.09% improvement in running speed.

## 4. Discussion

Accurate detection of individual geese and the segmentation of geese is a requirement for the development of precision animal husbandry and a feasible way to achieve a smart goose-breeding industry. Automated detection methods based on instance segmentation techniques can meet multiple needs of the livestock industry, while datasets from fully stocked models are more complex and informative. In this paper, the following topics were discussed.

### 4.1. Contribution and Effectiveness of the Proposed Method

Detection and segmentation were performed through computer vision and image-based processing methods. At a later stage, individual geese could be analyzed for behavior, body size, body condition, lameness, etc.; a flock could be counted and group activity analyzed. This method could effectively increase the scale of goose breeding and reduce production costs while effectively avoiding the spread of disease and improving the animal welfare of geese, etc. We built an instance segmentation model based on a goose dataset from free-range farms, aiming to accomplish two subtasks to assist other tasks. The dataset images had both single individual goose and dense geese activities and were disturbed by natural factors, such as vegetation shading, non-goose animals, water bodies, and debris. Compared to a captive breeding environment, the background environment was complex. Therefore, we proposed a more suitable high-precision algorithm with more robustness. The evaluation of the model showed that the query-based QueryPNet model could achieve an average accuracy of 0.963 for mAP@0.5. The model allowed for better extraction of goose body contour features. For example, despite the dense activity of the geese, it was also possible to detect and well-separate geese with the same body color, effectively avoiding miscalculation in later counts, etc.

This was a novel application of a query-based target detector and instance segmentation method to the livestock-farming industry. As far as we know, this is the first one. Therefore, the research in this paper fills this gap to a certain extent and provides a relevant reference for future research by other authors in this area, which is of practical significance.

### 4.2. Limitations and Future Developments

It should not be overlooked that the present study still has limitations.

First, the instance segmentation model was large and had more parameters compared to models for other tasks, e.g., pure target detection, semantic segmentation, classification, etc., although more subtasks could be implemented. The practical application is more difficult. The model proposed in this paper had an FPS of only about 7.3, which is suitable for deployment on high-calculus platforms and is unrealistic for edge devices. This is one of the directions of our future research: to reduce the model size and the number of training parameters to reach a level where the breeding condition can be monitored anytime and anywhere. By improving the model structure in the first stage and pruning, quantizing, and distilling the training model in the later stage, the model size can be reduced and unnecessary parameters eliminated while maintaining good performance.

Secondly, this collection of datasets only contained one species of goose. We should pay attention to the mixed breeding patterns of free-range farms and work on a piece of target detection to distinguish similar small geese classes under similar large geese classes. There are many more meat goose breeds that should be studied in as many subdivisions as conditions allow to avoid unnecessary misidentification. Therefore, we plan to continue to expand our dataset in the future and use transfer learning for better results in this aspect of livestock and poultry.

Finally, our model was based on the QueryInst model, which was proposed in 2021 and is an innovative model with novel research ideas. We improved the fusion of the neck module on this basis, and although good results could be obtained, there are still other, more superior methods. In future research, we plan to investigate instance segmentation models in more detail, continue to try to introduce more effective network modules, keep optimizing the structure of the QueryPNet model, and continue to improve the model effectiveness, especially in terms of speed.

In summary, this paper explored different algorithms to construct a model suitable for complex farming methods to achieve automated monitoring, for example, captive breeding for single species, captive breeding for mixed species, free-range breeding for mixed species, etc. We also aimed to deploy the model to embedded devices for large-scale practical applications in the future.

## 5. Conclusions

A robust goose detection and segmentation algorithm is essential in precision livestock-farming management. The implementation of an algorithm facilitates the detection of individual goose behavior and body size, the counting of geese, the extraction of contour lines, and the efficient and accurate analysis of goose breeding conditions. Therefore, to achieve accurate information acquisition in a real and complex free-range farm environment, we proposed a high-precision model. By reasonably designing the neck module of the model, more rich features were obtained for effective fusion. At the same time, the overall performance of the model was optimized to make the new model surpass in complexity, training cost, and speed based on ensuring high accuracy. Finally, experiments were conducted on our goose dataset, and the mAP@0.5 for both detection and segmentation reached 0.963.

For future research, we intend to explore in-depth in terms of accuracy and speed, aiming to achieve high-precision, real-time instance segmentation.

## Figures and Tables

**Figure 1 animals-12-02653-f001:**
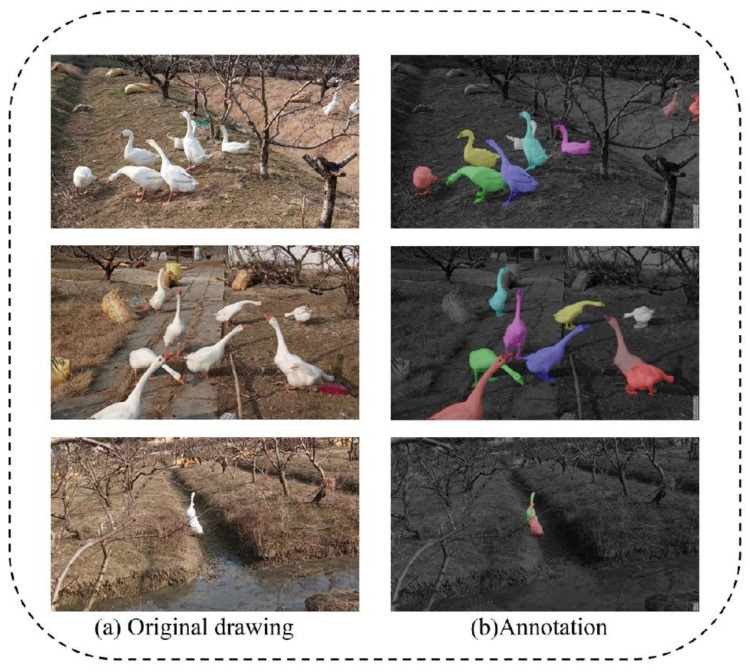
Goose breeding dataset and labeling schematic. (**a**) Original drawing represents the original dataset images. (**b**) Annotation represents a schematic representation of the dataset labels.

**Figure 2 animals-12-02653-f002:**
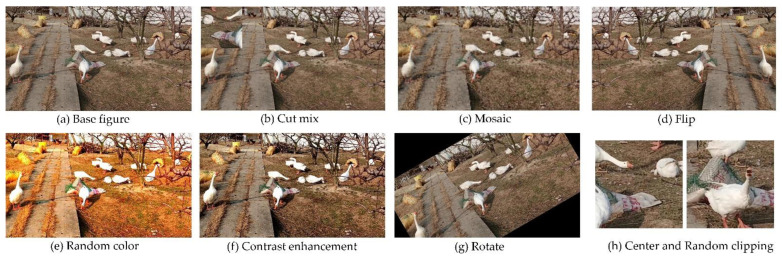
Effects of data enhancement. (**a**) Base figure represents the original image. (**b**) Cut mix represents the effect after cropping and combining the images. (**c**) Mosaic represents the effect of mosaic processing of the image. (**d**) Flip represents the effect after flipping the image. (**e**) Random color represents the effect of a random color transformation on the image. (**f**) Contrast enhancement represents the effect of a color contrast enhancement operation on the image. (**g**) Rotate represents the effect after rotating the image. (**h**) Center and Random clipping represents the effect after center cropping and random cropping of the image.

**Figure 3 animals-12-02653-f003:**
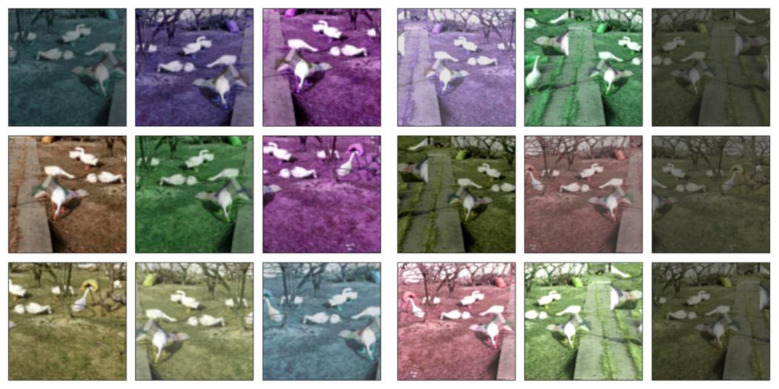
Convergence effect of data enhancement. For ease of understanding, the above image shows the visualization of our data after the enhancement operation.

**Figure 4 animals-12-02653-f004:**
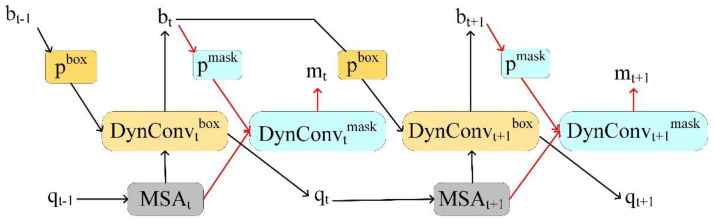
QueryInst with dynamic mask head. The red lines represent the mask branches. QueryInst consists of 6 iterative stages, t = {1, 2, 3, 4, 5, 6}, with 2 stages indicated in the figure.

**Figure 5 animals-12-02653-f005:**
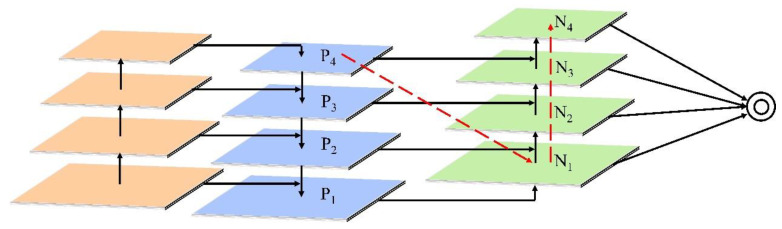
The main structure of the neck module. Red lines represent high-level features additionally augmented by bottom-up paths. P1~P4 indicate gradual down sampling with factor = 2. {N1, N2, N3, N4} correspond to the newly generated feature maps of {P1, P2, P3, P4}.

**Figure 6 animals-12-02653-f006:**
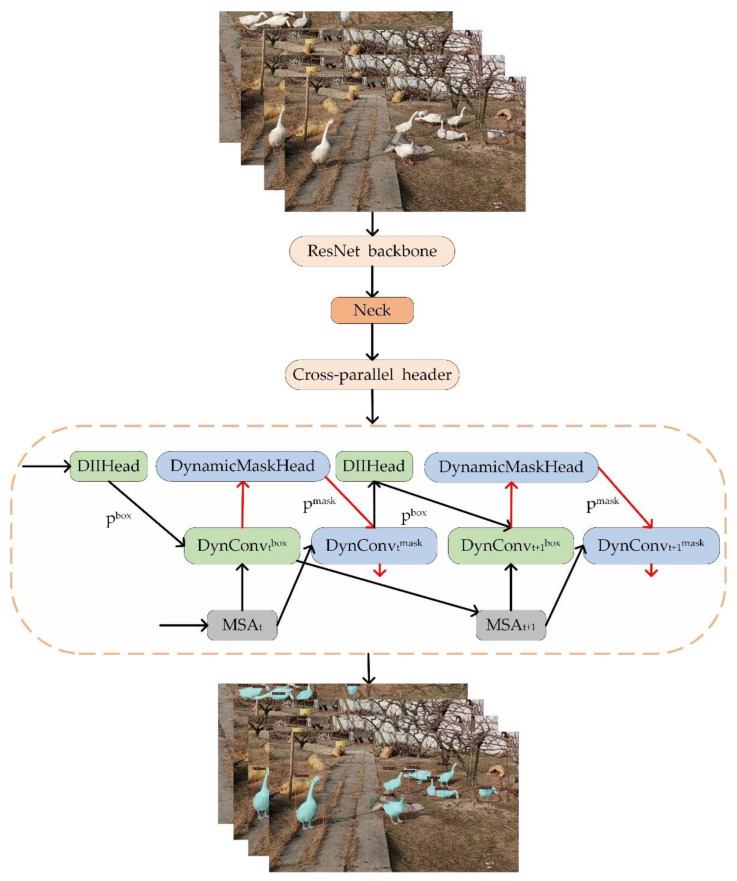
The QueryPNet model for the flock of geese. The red lines represent the mask branches. The model has a total of 5 cross-parallel headers. The design for the neck module is shown in Section 2.3.2.

**Figure 7 animals-12-02653-f007:**
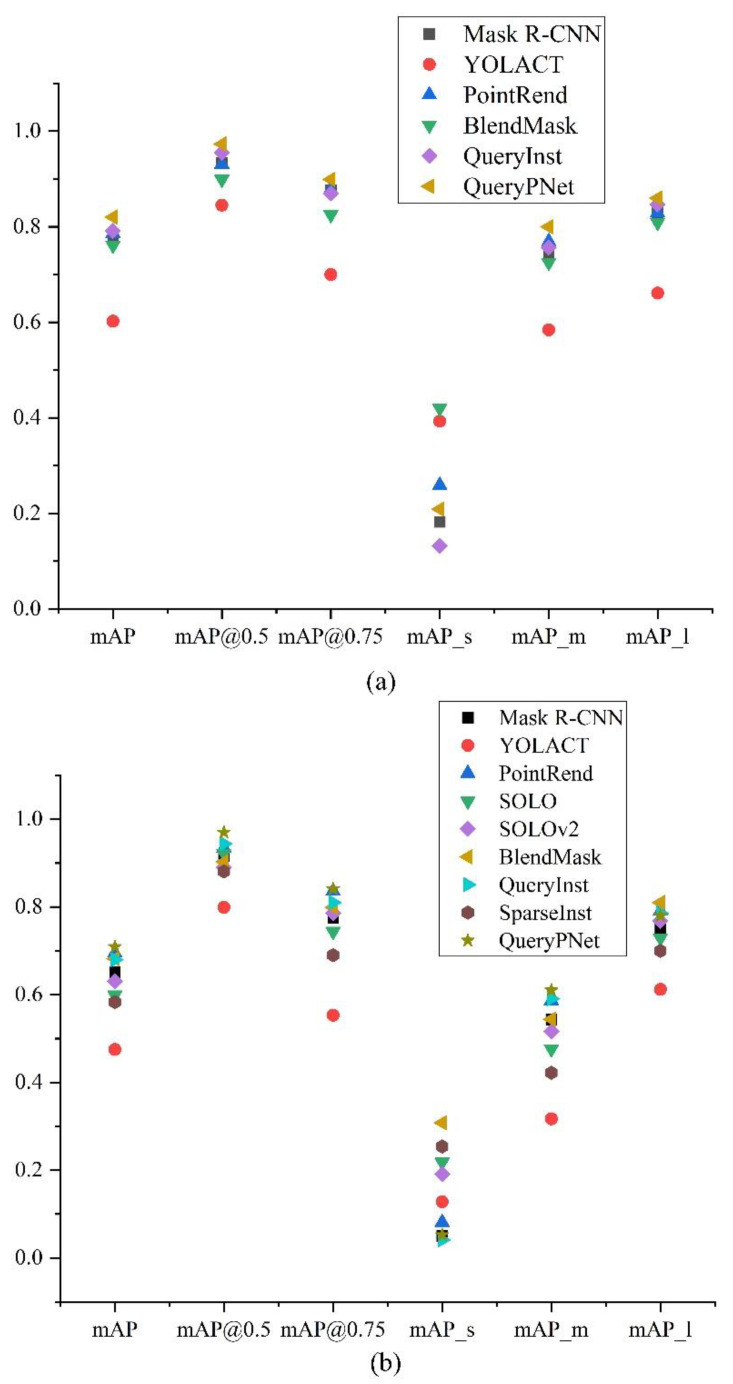
(**a**) Schematic diagram of the results for the detection of goose targets with different networks. (**b**) Schematic diagram of the results for the segmentation of individual goose instances with different networks.

**Figure 8 animals-12-02653-f008:**
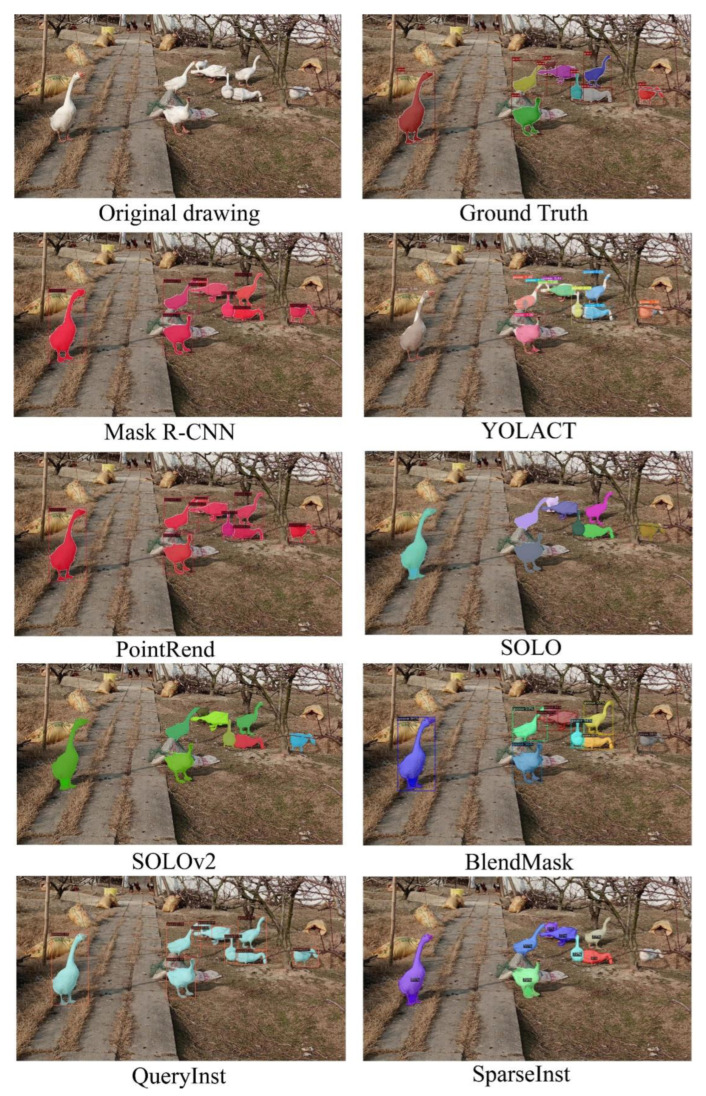
Different model effects display chart.

**Figure 9 animals-12-02653-f009:**
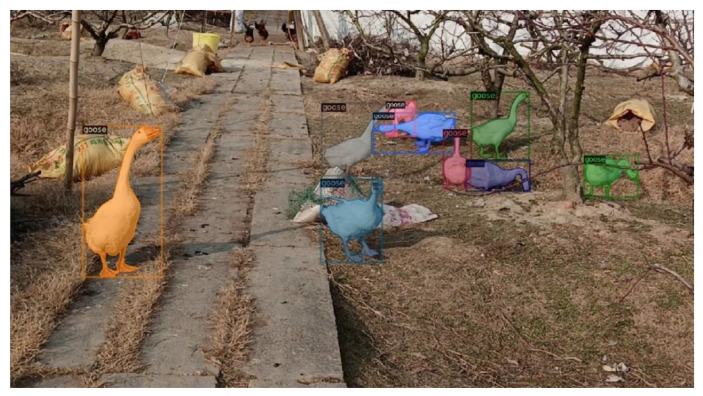
Effect demonstration diagram.

**Table 1 animals-12-02653-t001:** Data enhancement parameter settings.

Method	Setting
CutMix	Random
Mosaic	img_scale = (640, 640)
	prob = 1.0
RandomFlip	flip_ratio = [0.4, 0.4]
	direction = [‘horizontal’, ‘vertical’]
RandomColor	level = (0.255)
	prob = 0.8
Contrast enhancement	level = (0.90)
	prob = 0.8
Rotate	level = (0.90)
CenterCrop and RandomCrop	crop_size = (512, 256)
	prob = 0.8
Multi-scale training	height = 1333,
	weight = [480, 512, 544, 576, 608, 640, 672, 704, 736, 768, 800]

**Table 2 animals-12-02653-t002:** Software and hardware requirements.

Software	Type/Version	Hardware	Type/Version
Operating system	Ubuntu20.04	CPU	Intel(R) Xeon(R)
			Silver 4208 CPU @ 2.10 GHz
IDE	Pycharm	GPU	NVIDIA Corporation GV100 [TITAN V] (rev a1)
Python version	Python3.8	RAM	DDR4
Python library	Pytorch1.7.0	Hard disk	2 Terabytes

**Table 3 animals-12-02653-t003:** Description of the evaluation indicators.

Mean Average Precision	
IoU = 0.50:0.95	mAP
IoU = 0.50	mAP@0.5
IoU = 0.75	mAP@0.75
mAP Across Scales	
mAP for small objects: area < 32^2^	mAP_s
mAP for medium objects: 32^2^ < area < 96^2^	mAP_m
mAP for large objects: area > 32^2^	mAP_l

**Table 4 animals-12-02653-t004:** Results for target detection of geese with different networks.

Model	mAP	mAP@0.5	mAP@0.75	mAP_s	mAP_m	mAP_l
Mask	0.772	0.934	0.876	0.182	0.739	0.829
R-CNN						
YOLACT	0.602	0.845	0.700	0.393	0.584	0.661
PointRend	0.786	0.931	0.878	0.259	0.769	0.829
BlendMask	0.761	0.899	0.825	**0.420**	0.725	0.809
QueryInst	0.790	0.951	0.870	0.132	0.757	0.842
QueryPNet	**0.811**	**0.963**	**0.893**	0.209	**0.797**	**0.857**

The best results are in bold for each configuration.

**Table 5 animals-12-02653-t005:** Results for segmentation of individual goose instances with different networks.

Model	mAP	mAP@0.5	mAP@0.75	mAP_s	mAP_m	mAP_l
Mask	0.651	0.916	0.775	0.050	0.543	0.749
R-CNN						
YOLACT	0.475	0.799	0.553	0.128	0.317	0.612
PointRend	0.695	0.934	0.836	0.081	0.586	**0.791**
SOLO	0.599	0.922	0.744	0.218	0.476	0.729
SOLOv2	0.631	0.890	0.786	0.191	0.516	0.768
BlendMask	0.682	0.903	0.799	**0.308**	0.544	0.810
QueryInst	0.689	0.945	0.823	0.041	0.591	0.785
SparseInst	0.583	0.881	0.690	0.254	0.422	0.700
QueryPNet	**0.699**	**0.963**	**0.841**	0.046	**0.598**	0.780

The best results are in bold for each configuration.

**Table 6 animals-12-02653-t006:** Comparison of QueryInst and QueryPNet.

Model	Size (Pixel)	mAP^bbox^	mAP^segm^	FLOPs (G)	Params (M)	FPS
QueryInst	224	0.791	0.694	31.32	176.06	4.8
QueryPNet	224	**0.811**	**0.699**	**26.03**	**141.85**	**7.3**
▲		+0.02	+0.005	−5.29	−34.21	+2.5

“segm” denotes the segmentation of individual goose instances, and “bbox” denotes the target detection of the flock. FLOPs stands for floating-point operations per second and was used to measure the complexity of the model; the unit is G, which represents the number of floating-point operations per second in billions. Params is the total number of parameters to be trained in the network model; the unit is M, which means megabit. ▲ indicates change. The best results are in bold for each configuration. The FPS data were measured on a single TITAN V GPU with a batch size of 1.

## Data Availability

Inquiries regarding the data can be directed to the corresponding author.

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
