# Peer review of "Study of a QueryPNet Model for Accurate Detection and Segmentation of Goose Body Edge Contours"

_animals, 2022, doi:10.3390/ani12192653_

Round 1

Reviewer 1 Report

Very well written article. Few errors do not significantly affect its reception. The whole thing is logical.

The paper aims to prove that the proposed developed model of the automated detection method is feasible in the complex environment and can serve as a reference for the relevant development of the poultry industry, especially goose farming. I consider the paper's topic original since there is a lack of data considering using such a high-precision detection and segmentation model in poultry production.

I believe the conclusions are consistent with the presented arguments and address the main question posed.  

Below are some mistakes/things that need to be explained.

line 66: explain the abbreviation CNN, convolutional neural network?

line 82: rephrase it; "… of masks and is 20% faster than Mask R-CNN 20% faster than Mask;"

line 100 stated that "there is no instance segmentation research that has been applied related to poultry species." how about this paper? https://doi.org/10.1016/j.compag.2017.11.032

line 152: something wrong with numbers? starting from 4…

Reviewer 2 Report

This paper is developed within the framework of precision animal husbandry, aiming to explore the use of a computer vision technique, the novel QueryPNet, to detect, count and analyse the activity of geese. The Authors also mention that they have built the world’s first goose instance segmentation dataset.

The document is mostly well written, with some English language syntax issues, quite easy to read and to understand. It is organized as follows: Simple Summary, Abstract, a first Section containing a short Introduction, a second Section on the Dataset used, a third Section on the Method applied, a fourth Section on the Experiments, a fifht Section on the Discussion, a sixth Section including the Conclusions of the work, and finally the References used.

After a detailed review, I consider that this is an interesting paper. However, aiming to improve the quality of the work and without any intent to underrate neither its accuracy nor its contributions, I would like to make the following suggestions:

1. I might suggest to meke the title more specific, starting with: 'Proposal of a ..', 'Validation of a ...', 'Study of a...'

2.  About what is mentioned in Lines 58-60: ‘Non-invasive monitoring methods using sensors and cameras to acquire data and then process the data through computer vision have become a future research hotspot in precision animal husbandry (PLF) [9].’ Don’t the Authors think that this technology is a reality at the present?

3. According to the instructions for Authors: ‘We do not have strict formatting requirements, but all manuscripts must contain the required sections: Author Information, Abstract, Keywords, Introduction, Materials & Methods, Results, Conclusions, Figures and Tables with Captions, Funding Information, Author Contributions, Conflict of Interest and other Ethics Statements. Check the Journal Instructions for Authors for more details.’. I suggest you to reorganize the manuscript, so that it matches this structure.

4. Why do the Authors decide to use a query-based network model? What is its goodness in comparison with other architectures? How is other people using this architecture and for what purposes? What are the weaknesses of the proposed architecture in comparison with others?

5. I consider that after presenting Figure 6, an overall paragraph summary of the workflow of the architecture should be presented.

6. In section 4, first, add how many data you have before augmenting, and then, how many data you have after augmenting.

7. Which Python version was used?

8. In the case of the training setup, some specific configurations were stablished (learning rate, optimizer, weight decay rate) without adding any explanation or justification of the decisions made.

9. Could the proposed system, using the current training, be used for counting other animal species? If the environment is changed, could the system still be used for the same animal?

10. The Authors claim that they built the world’s first goose instance segmentation dataset. Is it public available to be used by others?

11. Do the Authors know if some similar system is currently being used in industrial settings?

12. Is it possible to estimate the expected savings? Could the Authors estimate the investment required to start running a system like this in an industry?

13. The mention of 'precision animal husbandry (PLF)' is confusing as the acronym does not match the name. Please check that acronym's usage in the document. I also consider that PLF is not an appropriate keyword, as it doesn’t help to filter. There are other terms which have the same acronym.

14. Check this and explain it (it is present in the Simple Summary and in the Abstract): mAP@0.5.

15. In Lines 37-40 it is mentioned: ‘In 2020, 76.39 million tonnes of pork, cattle, sheep and poultry meat will be produced in China, of which 23.61 million tonnes of poultry meat will be produced, an increase of 5.5% year-on-year, accounting for 30.9% of the total meat production.’ Is this a mistake in the year? We are currently in 2022. If the year is right, time verbs should be revised. A citation should be also added. From which source did you get this information?

15. Other potential issues to ve revised for language or consistency:

LINE NUMBER(S)        POTENTIAL ISSUE DETECTED

13, 21, 27 ...               Please check the correct use of 'Goose' or 'Geese'?
22                             'to avoid the influence of human,'
25-26                        The Authors mention: ‘which is effective in improving the traceability of geese, reducing costs and etc.’
42                             'an increase' redundancy
47                             'For this reason, realise intelligent precision farming can improve the scale and quality'
60                             'The accurate aquisition of livestock'
70                             'segmenting each example'
97 , 143 and 454        '1000' , '3247' , '1920×1080'   ->   '1,000' , '3,247' , '1,920×1,080'
99                             'agricultural farming'
101                           'Farming (PLF) farming'
119-120                     'captive farming could make the trained model has good robustness.'
132-134                      'so the geese raised have better quality meat, thus allowing maximum access to data on the health and welfare of the geese in terms of the environment and raising economic benefit'
149                            Use a more descriptive label for the figure
152-165                      Check list numbering
170                             'requirements such as blurred or do not contain geese'
186 and others             Check the use of italics for variables in the text
291, 293, 294               What do you mean by 'bin'?
Figure 2                       Revise the labels of Figure inserts. In the case of b you have a red line. Also try to add a space between the letter between brackets and the text. I consider that you also should try to improve the label of the Figure, adding the different descriptions you have in the image. For example: Figure 2. Effects of data enhancement. (a) base figure […].
313                              Use a more descriptive label for the figure
329                              A capital letter is missing: ‘the mask head is […]’.
358-359                       'a path aggregation network is used in the model and take high-level feature maps with rich segmentation information as an ad hoc input for better performance.'
369 and 616                 'Neck'
371                              'Then using bottom-up path enhancement'
376                              'use tiny fully connected layers to enhance predictions.'
397 and others              Check for mathematical expressions connected to the following word without spacing to separate them
434                              'for dice loss, is the loss function'
Figure 6                       Consider reducing font size to match the text size
Table 2:                        '2.10GHz' and 'Pyorch'  ->  '2.10 GHz' and 'Pytorch'
493                              'we choose priority to the mainstream networks in recent years'
499, 502 and 538           The asterisk note does not reference any insertion. Consider including the note into the table label.
Figure 7                        Increase figure quality, and increase text size
Figure 8                        It is difficult to compare the processed images with the Ground Truth because of their different backgrounds
Table 6                         'FLOPs(G)'  'Params(M)' , the terms in the parentheses are not defined or explained
560-561                        'This can effectively improve the traceability of geese, increase the scale of goose breeding to reduce production costs'
590-591                         'this collection of data set only labeled meat goose, and belong to the East Zhejiang white goose.'
615                                'this paper proposes an effective model that with high accuracy.'

Round 2

Reviewer 2 Report

Dear Authors, I thank you for your responses that I find very appropriate.

However, after revising them and the new version of the manuscript, I consider that it is very hard to verify the changes made, as you the Authors have neither indicated in which specific lines the changes were made, nor you had highlighted them or used any other colour in the text for the affected text pieces.

Because of that reason, I propose to make a new revision, to give you the opportunity of sorting out those issues.

Round 3

Reviewer 2 Report

After revising this new version of the manuscript, I consider that the paper can be accepted for publication.